# Epidemiology, Outcomes and Tolerability of Protracted Treatment of Nontuberculous Mycobacterial Infections at a Community Teaching Hospital in the Southeastern United States

**DOI:** 10.3390/antibiotics11121720

**Published:** 2022-11-29

**Authors:** Yuwei Vivian Tsai, Caroline Derrick, Ismaeel Yunusa, Sharon Weissman, Majdi N. Al-Hasan, Julie Ann Justo, Paul Brandon Bookstaver

**Affiliations:** 1Prisma Health-Midlands, Columbia, SC 29201, USA; 2Department of Clinical Pharmacy and Outcomes Sciences, University of South Carolina College of Pharmacy, Columbia, SC 29208, USA; 3Department of Internal Medicine, University of South Carolina School of Medicine, Columbia, SC 29209, USA

**Keywords:** nontuberculous mycobacteria, *Mycobacterium avium complex*, *Mycobacterium abscessus*, susceptibility

## Abstract

Nontuberculous mycobacterial (NTM) infections present a treatment challenge for clinicians and patients. There are limited data about current susceptibility patterns and treatment outcomes in U.S. adults. This was a 10-year, single-center, retrospective, observational cohort study of adults with a positive NTM culture and clinical suspicion of infection between 1 January 2010 and 30 June 2020. The primary objective was to identify predictors for favorable treatment outcomes. Key secondary objectives were characterization of NTM epidemiology, susceptibility profiles, and safety and tolerability of treatment, including the proportion of subjects with an antimicrobial change and the reasons for the change. Of 250 subjects diagnosed with NTM infection, the most prevalent NTM isolates were *Mycobacterium avium intracellulare complex* (66.8%) followed by *Mycobacterium abscessus* (17.6%). Antimicrobial susceptibility data were available for 52.4% of the cohort (45.8% slow growers; 54.2% rapid growers). Only 88 (35%) subjects received treatment with evaluable clinical outcomes. The proportion of subjects with a favorable outcome was 61.4%. More subjects in the unfavorable outcome group experienced a change in antimicrobial therapy (73.5% vs. 51.9%, *p* = 0.043). The most common reason for antimicrobial change was adverse drug events (*n* = 36, 67.9%). In the regression model, private insurance was associated with a favorable outcome, whereas having multiple antimicrobial changes was associated with an unfavorable outcome. The complexity of NTM treatment and high incidence of medication-related issues suggest the necessity of interdisciplinary collaboration to improve overall treatment outcomes in NTM infections.

## 1. Introduction

Nontuberculous mycobacteria (NTM) are common environmental habitants found in water sources and soil [1]. While NTM typically do not cause significant human diseases, they can become pathogenic in susceptible individuals [2,3,4,5,6,7]. The incidence of NTM infections in the United States (U.S.) appears to be the highest in the southeastern region, with variability in the overall geographical distribution of NTM species [8].

The management of NTM infections is heterogeneous, with a relatively low cure rate [9]. Current clinical practice guidelines recommend an in vitro susceptibility approach to treatment of NTM infections given the established correlation with clinical response [10,11], but limited evidence is available on current susceptibility patterns with most data reported outside of the U.S. [12,13,14,15]. Furthermore, limited studies have been conducted to examine the impact of antimicrobial regimens on NTM treatment outcomes and the burden these complex, multidrug regimens have on patients.

The primary objective of this present study was to identify predictors for favorable treatment outcomes associated with NTM infections in a Southeastern U.S. cohort. Key secondary objectives include (1) characterizing the local distribution and susceptibility patterns of NTM organisms and (2) evaluating the safety and tolerability of long-term treatment.

## 2. Results

Data from 292 adults with a positive culture for NTM species and clinical diagnosis for true infections were evaluated. A total of 13 and 29 subjects were excluded for concurrent MTB infection and monomicrobial culture with *M. gordonae*, respectively, resulting in 250 subjects included in the microbiological cohort. In establishing the treatment cohort, 152 subjects were further excluded due to the following: actively receiving therapy at the time of data analysis (*n* = 9), being followed outside of our system in the outpatient setting (*n* = 8), deceased before culture positivity (*n* = 11), lack of clinical documentation available for treatment outcome (*n* = 16), and did not receive treatment (*n* = 108). The most common reasons for not treating were asymptomatic disease (*n* = 48, 44.4%) and colonization/contamination (*n* = 25, 23.1%). A total of 88 subjects were included in the final analysis of treatment outcomes (Figure 1).

Table 1 shows the baseline characteristics of the microbiological cohort. The cohort consisted of 56.8% (*n* = 142) females and 66% (*n* = 165) non-Hispanic Caucasians with a median age of 67.4 years (IQR 24.1). The primary source of NTM infection was pulmonary (*n* = 197, 79.2%). The most common comorbidities included pulmonary disease (*n* = 125, 50.2%), history of smoking (*n* = 133, 53.4%), and immunocompromised state (*n* = 68, 27.3%) The predominate NTM organism observed was *Mycobacterium avium intracellulare complex* (MAC) (*n* = 167, 66.8%) followed by *Mycobacterium abscessus* (*M. abscessus*) (*n* = 44, 17.6%) (Figure 2). Baseline susceptibility data were only available for 52.4% (*n* = 131) of the cohort with 54.2% (*n* = 71) being rapid growers and 45.8% (*n* = 60) being slow growers. Overall, all the rapid growers remained highly susceptible to amikacin (*n*= 44, 90.9%). However, when excluding *M. abscessus* from the analysis, the susceptibility pattern for other rapid growers improved to a minimum of 70% for multiple antimicrobials including ciprofloxacin (*n* = 20, 74.1%), moxifloxacin (*n* = 21, 77.8%), linezolid (*n* = 26, 96.3%), and trimethoprim/sulfamethoxazole (SXT) (*n* = 19, 79.2%) (Table 2). Contrary to the rapid growers, MAC isolates remained highly susceptible to multiple antimicrobials including moxifloxacin (*n* = 51, 96.2%), clarithromycin (*n* = 53, 100%), linezolid (*n* = 41, 77.4%), intravenous (*n* = 51, 96.2%) and inhaled/liposomal (*n* = 53, 100%) amikacin (Table 3).

Within the treatment outcome cohort, the proportion of subjects with a favorable outcome was 61.4% (*n* = 54). More individuals in the unfavorable group were underweight (29.4% vs. 7.4%, *p* = 0.006) and uninsured/self-pay (29.4% vs. 13%, *p* = 0.057) compared to the favorable outcome group. Additionally, those with unfavorable outcome had higher rates of asthma (36.8% vs. 8%, *p* = 0.02) and prior history of MTB treatment (11.8% vs. 0%, *p* = 0.02) (Table 4). The median follow-up period was 270 days (IQR 318 days). Eighty-three percent (*n* = 73) of the treatment outcome cohort were initiated on at least three antimicrobials at baseline with macrolides (*n* = 76, 86.4%), ethambutol (*n* = 51, 58%), and rifamycin (*n* = 47, 53.4%) being the most common antimicrobials prescribed. About 60% (*n* = 53) of the treatment outcome cohort had a change in their antimicrobial therapy, and those with unfavorable outcome had a greater proportion of patients with a change in antimicrobial therapy (73.5% vs. 51.9%, *p* = 0.043). The most common reason for an antimicrobial change was due to adverse drug events (ADEs) (*n* = 36, 67.9%) (Table 5). In univariable analysis, private insurance was a predictor for a favorable outcome, whereas having more than five antimicrobial changes, being underweight, having a history of asthma, and having antimicrobial change due to intolerance were associated with an unfavorable outcome. In multivariable analysis, private insurance remained a significant protective factor with an odds ratio of 6.11 (95% Cl 1.12–33.29, *p* = 0.036); while having more than five antimicrobial therapy changes was associated with 81.3% higher risk for an unfavorable outcome (95% Cl 0.049–0.714, *p* = 0.014). (Table 6).

## 3. Discussion

The prevalence of NTM infections has steadily increased over the last few years. In a recent nationwide US Veterans Health Administration study, the incidence of NTM infections was 12.6 per 100,000 patient-years, with the highest occurrence observed in the southeastern US. The most common organisms observed were MAC followed by *M. chelonae-abscessus* group and *M. fortuitum* complex [8], which is a similar finding observed in the current study. MAC isolates remained highly susceptible to clarithromycin and amikacin in the current study perhaps reassurance as the recommended first-line treatment is a macrolide-based regimen with the addition of intravenous amikacin for fibrocavitary pulmonary disease and inhaled amikacin for salvage therapy [10,11]. In comparison to MAC, the susceptibility profile of multiple antimicrobials for *M. abscessus* was relatively poor except for amikacin and tigecycline, which is also consistent with other reports [12,13]. Additionally, the wider spread of MICs of antimicrobials to rapid growers as compared to slow growers suggests likelihood of reduced target attainment. While data on pharmacodynamics of these antimicrobials are limited, synergy is observed and supported by clinical outcomes [16,17,18].

Since NTM organisms are ubiquitous, determination of their clinical significance is pertinent to rule out specimen contamination or colonization prior to initiation of long-term, complex antimicrobial course. In our study, 52.4% of the subjects did not receive therapy due to asymptomatic cases (44.4%) and culture colonization/contamination (23.1%) being the most common reasons. Previous studies have cited variable rates of 15 to 76.6% where treatment was held with watchful waiting for NTM pulmonary disease [9,19,20,21,22]. In patients meeting the diagnostic criteria for NTM infections, initiation of therapy is often recommended, especially among those with factors associated with relatively poor prognosis [10,22,23]. In our study, those with an unfavorable outcome were more likely to be underweight, and have a history of asthma and prior MTB treatment. The higher rate of asthma history in unfavorable outcome group is notable since asthma is not considered as a restrictive pulmonary disease like cavitation or chronic obstructive pulmonary disease. Prior history of MTB could also influence treatment outcomes due to overlapping use of antimicrobials such as rifamycins and ethambutol potentially influencing susceptibility patterns.

Despite the use of long-term, multiple antimicrobials, the cure rate for NTM infections is often low. In a recent multicenter study of pulmonary NTM infections, the proportion of subjects with symptom improvement at 3 months was 45%, and the rate for favorable treatment outcome was only 56.6% [10]. The present study also found a similar result of 61.4% of patients with a favorable outcome. There are a few reasons that can explain the low cure rate of NTM infections. The prolonged use of multiple antimicrobials comes with the inherent risk for ADEs, which could be more apparent in NTM infected population since epidemiological data suggested that susceptible individuals are typically observed in those who are immunocompromised, older, and slender in body habitus [2,3,4,5,6]. In our study, 60.2% of the treatment outcome cohort had at least one antimicrobial change, with ADEs being the most common reason, which is consistent with other studies [24,25]. Another potential reason explaining the low cure rate for NTM infections is difficulty in medication adherence. While not directly measured in the present study, the known complexity of NTM regimens including multiple delivery modalities, multi-daily dosing, and barriers to medication access complicate medication adherence. Interdisciplinary collaboration among clinicians, pharmacists, and other care coordinators is vital for optimizing treatment outcomes. Recently, Brizzi and colleagues reported a successful implementation of a pharmacist-driven antiretroviral (ART) stewardship and transition of care (TOC) program for persons with HIV. Medication error rates associated with ART or opportunistic infection medications and 30-day all-cause readmission rate decreased significantly from 17% to 6% and 27% to 12% with the implementation of ART stewardship, respectively. Additionally, the rate for linkage to care also increased significantly from 78% to 92% [26]. Similar to patients receiving ART, patients with NTM infections face a complex treatment course. Parenteral therapies require close monitoring and line-related complications are difficult to measure; however, they are of significant clinical relevance. For example, aminoglycosides require renal, vestibular, and ototoxicity monitoring while oral therapies discussed here such as ethambutol and rifampin require ocular examinations and liver and renal function tests, respectively. As demonstrated in our study, medication-related issues were not only limited to ADEs but the need to navigate through drug access or administration route concerns are also some frequent reasons for interruptions in antibiotic therapy. Therefore, an interprofessional collaboration between physicians, pharmacists, nurses, and case managers is pertinent to not only ensuring safe and appropriate NTM treatment but also optimizing medication adherence, and thus likely influencing treatment response.

In our evaluation, the only factor in both univariable and multivariable regression analyses associated with favorable outcomes was having private insurance. Adequate insurance coverage may allow for expanded access to more alternative and expensive agents (e.g., tedizolid, omadacycline), which is pertinent in managing NTM infections where changes in antimicrobial therapy are often observed. In addition, the only harmful factor identified in our study was having more than 5 antimicrobial adjustments, which highlights the intolerability and potential barriers to NTM regimens that often leads to treatment discontinuation.

Limitations to this retrospective study include: (1) the cohort consisted predominately of pulmonary NTM, thereby generalizability to other sources of infections is limited; (2) difficulty in correlating antimicrobial-specific ADEs in the outpatient setting due to multiple antimicrobials in each regimen; (3) lack of data evaluating the patient adherence contribution to treatment success; (4) the use of physician-guided cessation of therapy due to clinical improvement to define favorable treatment outcome. While clinical practice guidelines suggest the use of sputum conversion in pulmonary NTM as guidance for treatment duration, obtainment of follow-up cultures was not consistently available, and a specialist team may often discontinue therapy in the presence of persistent colonization, especially if the patient has received a prolonged course with a favorable clinical response; (5) analysis of all NTM organisms may have influenced study outcomes since certain NTM species, e.g., *M. abscessus* require different treatment regimens and have varying prognoses; and; (6) the wide confidence interval observed in multivariable regression analysis might be driven by the relatively few number of events and sample size [27].

## 4. Materials and Methods

This was a retrospective, observational cohort study at a single healthcare system in the Southeastern US. Patients with a positive NTM culture between 1 January 2010 and 30 June 2020 were eligible for study enrollment. Patients were included if they were at least 18 years of age or older with clinical suspicion of NTM infections. The enrolled population was further stratified into three groups for data analysis. The microbiological cohort included all patients with a positive NTM culture except those (1) with concurrent *Mycobacterium tuberculosis* (MTB) infection and (2) monomicrobial culture positive for *M. gordonae* as it is typically considered as an environmental contaminant. The treatment outcome cohort included all patients in the microbiological cohort except those (1) actively receiving treatment during the study period, (2) followed outside our system in the outpatient setting, (3) deceased prior to culture positivity, and (4) lack of clinical documentation to determine study outcome. Finally, the treated cohort included all patients in the treatment outcome cohort except those who did not receive antimicrobial therapy at the treating physician’s discretion (Figure 1).

A favorable treatment outcome was defined as physician-guided cessation of therapy due to clinical improvement. An unfavorable treatment outcome was defined as mortality or transitioned to palliative or hospice care while on therapy, or termination of treatment due to antimicrobial intolerability or lack of clinical improvement. Comorbidities were recorded at the time of NTM diagnosis. Co-infections are defined as any positive non-NTM cultures at the time of NTM diagnosis that were considered true infections per the physician’s discretion. The use of an immunosuppressant was defined as receipt of systemic steroids at a dose equivalent to ≥15 mg/day prednisone for at least 1 month, TNF-alpha inhibitors, or other non-corticosteroid immunosuppressive medications. Source control was defined at physician’s discretion per clinical documentation. An immunocompromised state was defined as a history of malignancy, solid organ transplant, and HIV. Cavitary disease was defined as a radiographic description of the presence of cavitation. Antimicrobial regimen change evaluation, the clinical note pertaining to treating NTM infections for each enrolled patient was examined from initiation until therapy discontinuation. Antimicrobial regimen changes were documented, including the name of the agent(s), the reason for the change, and the interventions made, if any. Reasons for antimicrobial change were placed into nine categories, including the following: drug–drug interaction, antibiotic susceptibility, disease status, treatment optimization, ADEs, drug access, administration issue, other, and nonspecific/unclear. Within disease status, escalation and de-escalation of therapy were noted. Treatment optimization was defined as adding antimicrobial agents to complete the existing regimen per provider.

Data were presented as proportions, mean (standard deviations [SD]), or medians (interquartile range [IQR]). Categorical variables were summarized with Chi-squared test or Fisher exact test as appropriate. Continuous variables were summarized with student *t-*test or Mann–Whitney test as appropriate. Susceptibility data were interpreted using Clinical and Laboratory Standards Institute (CLSI) M62 Performance Standards for Susceptibility Testing of Mycobacteria, *Nocardia* spp., and Other Aerobic Actinomycetes 1st Edition. Only the first culture per patient during the study period was considered for susceptibility distribution. The most comprehensive susceptibility testing was reported. Logistic regression models were fitted to identify factors predicting favorable treatment outcome. Variables found significant in the bivariate analysis with *p* < 0.05 or based on clinical decision as a confounder for treatment outcome were included in the final multivariable model. All tests were two-sided, and a *p*-value of < 0.05 was considered statistically significant. Effect estimates were summarized as odds ratios with a 95% confidence interval. Analyses were conducted using Stata statistical software version 17.0 (StataCorp).

## 5. Conclusions

In this single-center, retrospective evaluation of 10-year treatment experience of NTM infections, MAC was the most prevalent isolate, followed by *M. abscessus*. Antimicrobial susceptibility pattern predominately limits treatment options for *M. abscessus*. Nearly two-thirds of the cohort required at least one antimicrobial change, which was significantly more common among those with unfavorable outcomes. While ADEs was the most common reason for antimicrobial change, undesirability of route of administration, financial challenges, and susceptibility patterns also contributed significantly to therapy interruptions or changes. Factors predicting a favorable outcome was having private insurance while having more than five antimicrobial changes was associated with an increased risk for treatment failure. To our knowledge, this is the first report of characterizing susceptibility patterns of NTM organisms in the Southeastern U.S. and associated treatment outcomes, providing additional evidence for patient specific decision-making. Furthermore, the complexity of NTM treatment and high incidence of medication-related issues suggest the necessity of interprofessional collaboration to improve overall treatment outcomes of NTM infections.

## Figures and Tables

**Figure 1 antibiotics-11-01720-f001:**
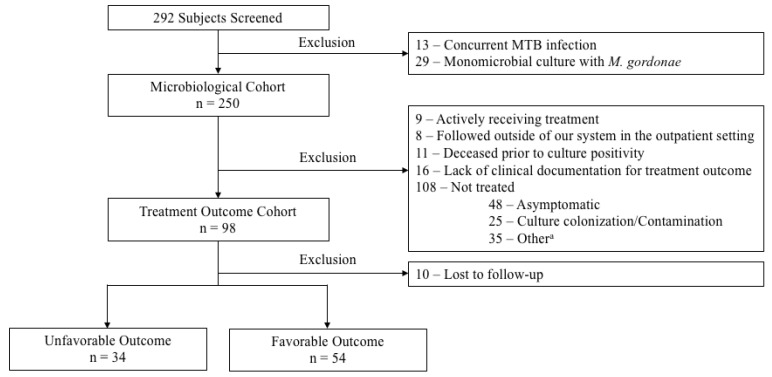
Study Cohort. ^a^ Other reasons for not treating included the following: lost to follow-up (*n* = 10), source controlled (*n* = 5), patient refused treatment (*n* = 5), hospice/palliative care (*n* = 2), risks outweigh benefits per clinicians’ recommendation (*n* = 2), pregnancy (*n* = 1), and unclear/undetermined (*n* = 10).

**Figure 2 antibiotics-11-01720-f002:**
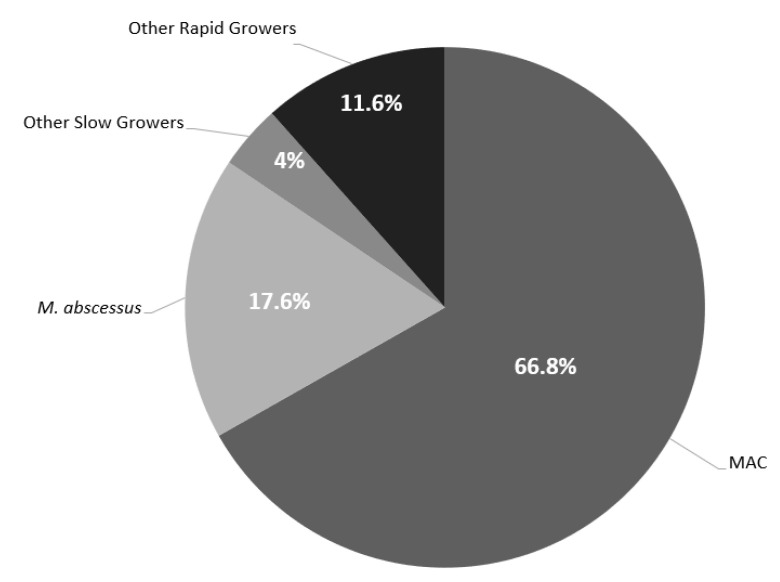
Distribution of NTM Isolates. Other Rapid Growers included: *M. chelonae* (*n* = 5), Other Rapid Growers included: *M. chelonae* (*n* = 5), *M. fortuitum* (*n* = 16), *M. mucogenicum* (*n* = 3), *M. brisbanense* (*n* = 1), *M. goodii* (*n* = 2), *M. smegmatis* (*n* = 1), and *M. immunogenum* (*n* = 1). Other Slow Growers included: *M. kansasii* (*n* = 2), *M. marinum* (*n* = 2), *M. neoarum* (*n* = 1), *M. haemophilum* (*n* = 2), *M. simiae/interjectum* (*n* = 1), *M. lentiflavum* (*n* = 1), and *M. asiaticum* (*n* = 1).

**Table 1 antibiotics-11-01720-t001:** Baseline Characteristics of Microbiological Cohort.

	All (*n* = 250)	MAC (*n* = 167)	*M. abscessus* (*n* = 44)	Other Slow Growers ^a^ (*n* = 10)	Other Rapid Growers ^b^ (*n* = 29)
Age (y), median (IQR)	67.4 (24.1)	68.4 (21.9)	65.7 (17.8)	56.5 (21.9)	66 (29.5)
Male, *n* (%)	108/250 (43.2)	70/167 (41.9)	21/44 (47.7)	8/10 (80)	9/29 (31)
BMI (kg/m^2^), *n* (%)					
Underweight, (<18.5)	29/250 (11.6)	18/167 (10.8)	7/44 (15.9)	2/10 (20)	2/29 (6.9)
Normal weight, (18.5–24.9)	92/250 (36.8)	67/167 (40.1)	15/44 (34.1)	5/10 (50)	5/29 (17.2)
Overweight, (25–29.9)	44/250 (17.6)	27/167 (16.2)	8/44 (18.2)	3/10 (30)	6/29 (20.7)
Obese, (30–39.9)	34/250 (13.6)	20/167 (12)	7/44 (15.9)	0 (0)	7/29 (24.1)
Severely obese, (≥40)	13/250 (5.2)	9/167 (5.4)	1/44 (2.3)	0 (0)	3/29 (10.3)
Race/Ethnicity, *n* (%)					
Non-Hispanic white	165/250 (66)	114/167 (68.3)	28/44 (63.6)	4/10 (40)	19/29 (65.5)
Non-Hispanic black	80/250 (32)	51/167 (30.5)	13/44 (29.5)	6/10 (60)	10/29 (34.5)
Hispanic	3/250 (1.2)	2/167 (1.2)	1/44 (2.3)	0 (0)	0 (0)
Unknown	2/250 (0.8)	0 (0)	2/44 (4.5)	0 (0)	0 (0)
Insurance, *n* (%)					
Private	33/250 (17.6)	24/167 (14.4)	12/44 (27.3)	3/10 (30)	5/29 (17.2)
Medicare	122/250 (48.8)	84/167 (50.3)	19/44 (43.2)	4/10 (40)	15/29 (51.7)
Medicaid	25/250 (10)	18/167 (10.8)	2/44 (4.5)	2/10 (20)	3/29 (10.3)
Uninsured	59/250 (23.6)	41/167 (24.6)	11/44 (25)	1/10 (10)	6/29 (20.7)
Type of NTM, *n* (%)					
Pulmonary	197/250 (79.2)	150/167 (89.8)	26/44 (59.1)	6/10 (50)	15/29 (51.7)
Extrapulmonary	50/250 (20)	14/167 (8.4)	18/44 (40.9)	4/10 (40)	14/29 (48.3)
Tissue	30/50 (60)	7/14 (50)	8/18 (44.4)	4/4 (100)	11/14 (78.6)
Blood	11/50 (22)	7/14 (50)	4/18 (22.2)	0 (0)	0 (0)
Other	9/50 (18)	0 (0)	6/18 (33.3)	0 (0)	3/14 (21.4)
Mixed	3/250 (1.2)	3/167 (1.8)	0 (0)	1/10 (10)	0 (0)
Co-infection, *n* (%)	52/250 (20.8)	36/167 (21.6)	8/44 (18.2)	3/10 (30)	5/29 (17.2)
Immunosuppressant, *n* (%)	34/250 (13.6)	23/167 (13.8)	6/44 (13.6)	2/10 (20)	3/29 (10.3)
Chronic Comorbidities, *n* (%) ^c^	194/249 (77.9)	134/167 (80.2)	33/43 (76.7)	5/10 (50)	22/29 (75.9)
DM, *n* (%)	35/249 (14.1)	19/167 (11.4)	8/43 (18.6)	1/10 (10)	7/29 (24.1)
CKD, *n* (%)	19/249 (7.6)	13/167 (7.8)	2/43 (4.7)	1/10 (10)	3/29 (10.3)
Liver disease, *n* (%)	6/249 (2.4)	3/167 (1.8)	3/43 (7.0)	0 (0)	0 (0)
Pulmonary disease, *n* (%)	125/249 (50.2)	88/167 (52.4)	22/43 (51.2)	1/10 (10)	14/29 (48.3)
COPD	37/125 (29.6)	24/88 (27.3)	7/22 (31.8)	0 (0)	6/14 (42.9)
Bronchitis	13/125 (10.4)	12/88 (13.6)	2/22 (9.1)	0 (0)	0 (0)
Emphysema	6/125 (4.8)	2/88 (2.3)	2/22 (9.1)	0 (0)	2/14 (14.3)
CF	4/125 (3.2)	1/88 (1.1)	3/22 (13.6)	0 (0)	0 (0)
Lung cancer	3/125 (2.4)	2/88 (2.3)	1/22 (4.5)	0 (0)	0 (0)
Bronchiectasis	33/125 (26.4)	20/88 (22.7)	6/22 (27.3)	1/1 (100)	6/14 (42.9)
Asthma	23/125 (18.4)	15/88 (17)	4/22 (18.2)	1/1 (100)	3/14 (21.4)
Interstitial lung disease	7/125 (5.6)	6/88 (6.8)	0 (0)	0 (0)	1/14 (7.1)
Nodules/Masses	15/125 (12)	15/88 (17)	0 (0)	0 (0)	0 (0)
Empyema	2/125 (1.6)	2/88 (2.3)	0 (0)	0 (0)	0 (0)
Fibrosis	4/125 (3.2)	3/88 (3.4)	0 (0)	0 (0)	1/14 (7.1)
Immunocompromised, *n* (%) ^d^	68/249 (27.3)	50/167 (29.9)	8/43 (18.6)	4/10 (40)	6/29 (20.7)
Prior NTM infection, *n* (%)	33/249 (13.3)	21/167 (12.6)	6/43 (14.0)	1/10 (10	5/29(17.2)
Treated	19/33 (57.6)	12/21 (57.1)	5/6 (83.3)	0 (0)	2/5 (40)
Not treated	14/33 (42.4)	9/21 (42.9)	1/6 (16.7)	1/1 (100)	3/5 (60)
Prior MTB treatment, *n* (%)	6/249 (2.4)	4/167 (2.4)	1/43 (2.3)	1/10 (10)	0 (0)
Smoking, *n* (%)	133/249 (53.4)	97/167 (58)	18/43 (41.9)	4/10 (40)	14/29 (48.3)

^a^ Does not include MAC; ^b^ Does not include *M. abscessus*; ^c^ Missing data for 1 individual in All cohort (*n* = 249) and missing data for 1 individual in *M. abscessus* cohort (*n* = 43); ^d^ Includes malignancy, history of prior transplant, and human immunodeficiency virus (HIV); Abbreviations: BMI, body mass index; CF, cystic fibrosis; CKD, chronic kidney disease; COPD, chronic obstructive pulmonary disease; DM, diabetes mellitus; IQR, interquartile range; MAC, *Mycobacterium avium intracellulare* complex; MTB, *Mycobacterium tuberculosis*; NTM, nontuberculous mycobacteria.

**Table 2 antibiotics-11-01720-t002:** MIC Distribution of Select Antimicrobials against Clinical Isolates of Rapid Growers collected from 2010 to 2020.

Antimicrobials (No. Isolates Tested)	No. of Isolates and Cumulative % of Inhibited at MIC of:															MIC (mg/L)	
	0.016	0.03	0.06	0.12	0.25	0.5	1	2	4	8	16	32	64	128	256	50%	90%
Ciprofloxacin																	
*M. abscessus* (44)									4	40						8	8
									9.1%	100%							
*Other* (27)				5	7	7	1	1	4	2						≤1	≥4
				18.5%	44%	70.4%	74.1%	75%	92.6%	100%							
Total (71)				5	7	7	1	1	8	42						8	8
				7.0%	16.9%	26.8%	28.2%	29.6%	40.8%	100%							
**Moxifloxacin**																	
*M. abscess* (44)									2	4	37	1				16	16
									4.5%	13.6%	97.7%	100%					
*Other* (27)					18	1	2		3	2	1					0.25	8
					66.7%	70.4%	77.8%		88.9%	96.3%	100%						
Total (71)					18	1	2		5	6	38	1				16	16
					25.4%	26.8%	29.6%		36.6%	45.1%	98.6%	100%					
**Cefoxitin**																	
*M. abscessus* (44)										1	5	29	7		2	32	≥64
										2.3%	13.6%	79.5%	95.5%		100%		
*Other* (27)										1	3	8	7	2	6	≥64	256
										3.7	18.5	44.4	70.4	77.8	100%		
Total (71)										2	8	37	14	2	8	32	256
										2.8%	14.1%	66.2%	65.9%	88.7	100%		
**Doxycycline**																	
*M. abscessus* (44)										2	2	40				32	32
										4.5%	9.1%	100%					
*Other* (27)				5		1	2		2	3	4	10				16	32
				18.5%		22.2%	29.6%		37%	48.1%	63%	100%					
Total (71)				5		1	2		2	5	6	50				32	32
				7.0%		8.5%	11.3%		14.1%	21.1%	29.6%	100%					
**Tigecycline ^a^**																	
*M. abscessus* (27)			5	11	9	1		1								0.12	0.25
			18.5%	59.35	92.6%	96.3%		100%									
*Other* (19)	7	4	6	2												0.03	0.12
	36.8%	57.9%	89.5%	100%													
Total (46)	7	4	11	13	9	1		1								0.12	0.25
	15.2%	23.9%	47.8%	76.1%	95.7%	97.8%		100%									
**Clarithromycin**																	
*M. abscessus* (44)				1	3	4	7			5	9	15				16	32
				2.3%	9.1%	18.2%	34.1%			45.5%	65.9%	100%					
*Other* (25)			1	3	2	3	2			6	5	3				≥8	32
			4%	16%	24%	36%	56%			64%	84%	100%					
Total (69)			1	4	5	7	9			11	14	18				≥8	32
			1.4%	7.2%	14.5%	24.6%	37.7%			53.6%	73.9%	100%					
**Linezolid**																	
*M. abscessus* (44)									4	15	18	7				16	32
									9.1%	43.2%	84.1%	100%					
*Other* (27)							5	5	6	10	1					4	≤8
							18.5%	37%	59.3%	96.3%	100%						
Total (71)							5	5	10	25	19	7				≤8	16
							7%	14.1	28.2%	63.4%	90.1%	100%					
**Imipenem**																	
*M. abscessus* (41)										12	24	5				16	32
										29.3%	87.8%	100%					
*Other* (27)							1	4	9	6	2	5				≤4	32
							3.7%	18.5%	51.9%	74.1%	81.5%	100%					
Total (68)							1	4	9	18	26	10				16	32
							1.5%	7.4%	20.6%	47.1%	85.3%	100%					
**Amikacin**																	
*M. abscessus* (44)										1	37	5	1			≤16	32
										2.3%	86.4%	97.7%	100%				
*Other* (22)							17	3		1	1					1	2
							77.3%	90.9%		95.5%	100%						
Total (66)							17	3		2	38	5	1			≤16	≤16
							25.8%	30.3%		33.3%	90.9%	98.5%	100%				
**Tobramycin**																	
*M. chelonae* (5)							2	3								NA	NA
							40%	100%									
**Minocycline ^b^**																	
*M. abscessus* (43)									4	21	18					8	16
									9.3%	58.1%	100%						
*Other* (26)							8	2	2	7	7					4	16
							30.8%	38.5%	84.6%	73.1%	100%						
Total (69)							8	2	6	28	25					8	16
							11.6%	14.5%	23.2%	63.8%	100%						
**SXT**																	
***M. abscessus* (44)**									2	13	29					8	16
									4.5%	34.1%	100%						
*Other* (27)					7	4	2	6	2	2	3					1	≥4
					25.9%	40.7%	48.1%	70.4%	77.8%	88.9%	100%						
Total (71)					7	4	2	6	4	15	32					16	16
					9.9%	15.5%	18.3%	26.8%	32.4%	53.5%	100%						

Based on CLSI breakpoint from M62 Performance Standards for Susceptibility Testing of Mycobacteria, *Nocardia* spp., and Other Aerobic Actinomycetes; MIC_50_ and MIC_90_ were not calculated if <10 isolates; Green: susceptible; Yellow: intermediate; Red: Resistant; ^a,b^ CLSI breakpoint not available.

**Table 3 antibiotics-11-01720-t003:** MIC Distribution of Select Antimicrobials against Clinical Isolates of Slow Growers collected from 2010 to 2020.

Antimicrobial (No. Isolates Tested)	No. of Isolates and Cumulative % of Inhibited at MIC of:												MIC (mg/L)	
	0.06	0.12	0.25	0.5	1	2	4	8	16	32	64	128	50%	90%
Ciprofloxacin														
*MAC* (11) ^a^		1	1		1	3	2	3					2	8
		9.1%	18.2%		27.3%	54.5%	72.7%	100%						
*Other* (3)		1	1	1									NA	NA
		33.3%	66.7%	100%										
**Moxifloxacin**														
*MAC* (53)		7	14	26	4			1	1				0.5	≤1
		13.2%	39.6%	88.7%	96.2%			98.1%	100%					
*Other* (3)		1	1	1									NA	NA
		33.3%	66.7%	100%										
**Clarithromycin**														
*MAC* (53)	9	10	15	9	7	2		1					0.25	1
	17%	35.8%	64.2%	81.1%	94.3%	98.1%		100%						
*M. kansasii* (2)				1			1						NA	NA
				50%			100%							
*Other* (5)	3	1							1				NA	NA
	60%	80%							100%					
**Linezolid**														
*MAC* (53)					6	10	10	15	9	2	1		≤8	16
					11.3%	30.2%	49.1%	77.4%	94.3%	96.2%	100%			
*Other* (3)					3								NA	NA
					100%									
**Amikacin intravenous**														
*MAC* (53)					7	12	19	11	2	1	1		4	8
					13.2%	35.8%	71.7%	92.5%	96.2%	98.1%	100%			
*Other* (5)					5								NA	NA
					100%									
**Amikacin liposomal or inhaled**														
*MAC* (53)					7	12	19	11	2	1	1		4	8
					13.2%	35.8%	71.7%	92.5%	96.2%	98.1%	100%			
**Doxycycline**														
*Other* (3)		2				1							NA	NA
		66.7%				100%								
**Rifampin**														
*M.kansasii* (2)		2											NA	NA
		100%												
*Other* (3)		3											NA	NA
		100%												
**Ethambutol ^b^**														
*Other* (3)				2					1				NA	NA
				66.7%					100%					
**SXT**														
*Other* (4)		1	1	1				1					NA	NA
		25%	50%	75%				100%						

Based on CLSI breakpoint from M62 Performance Standards for Susceptibility Testing of Mycobacteria, *Nocardia* spp., and Other Aerobic Actinomycetes; MIC_50_ and MIC_90_ were not calculated if <10 isolates; Green: susceptible; Yellow: intermediate; Red: Resistant; ^a,b^ CLSI breakpoint not available.

**Table 4 antibiotics-11-01720-t004:** Baseline Characteristics of Treatment Outcome Cohort.

	All (*n* = 88)	Unfavorable (*n* = 34)	Favorable (*n* = 54)	*p*-Value
Age (y) median (IQR)	66.8 ± 27	66.9 ± 32.7	66.8 ± 15.7	0.48
Male, *n* (%)	35/88 (39.8)	11/34 (32.4)	24/54 (44.4)	0.26
Body Mass Index (kg/m^2^), *n* (%)				
Underweight, (<18.5)	14/88 (15.9)	10/34 (29.4)	4/54 (7.4)	**0.006**
Normal weight, (18.5–24.9)	32/88 (36.4)	11/34 (32.4)	21/54 (38.9)	0.54
Overweight, (25–29.9)	14/88 (15.9)	3/34 (8.8)	11/54 (20.4)	0.15
Obese, (30–39.9)	10/88 (11.4)	5/34 (14.7)	5/54 (9.3)	0.50
Severely obese, (≥40)	4/88 (4.5)	1/34 (2.9)	3/54 (5.6)	1.00
Race/Ethnicity, *n* (%)				
Non-Hispanic white	64/88 (72.7)	24/34 (70.6)	40/54 (74.1)	0.72
Non-Hispanic black	23/88 (26.1)	10/34 (29.4)	13/54 (24.1)	0.58
Hispanic	1/88 (1.1)	0 (0)	1/54 (1.9)	1.00
Insurance, *n* (%)				
Private	17/88 (19.3)	2/34 (5.9)	15/54 (27.8)	**0.011**
Medicare	48/88 (54.5)	18/34 (52.9)	30/54 (55.6)	0.81
Medicaid	6/88(6.8)	4/34 (11.8)	2/54 (3.7)	0.20
Uninsured	17/88 (19.3)	10/34 (29.4)	7/54 (13)	**0.057**
Type of NTM, *n* (%)				
Pulmonary	61/88 (69.3)	24/34 (70.6)	37/54 (68.5)	0.84
Extrapulmonary	25/88 (28.4)	9/34 (26.5)	16/54 (29.6)	0.75
Mixed	2/88 (2.3)	1/34 (2.9)	1/54 (2.9)	1.00
Co-infection, *n* (%)	15/88 (17)	5/34 (14.7)	10/54 (18.5)	0.64
On immunosuppressants, *n* (%)	13/88 (14.8)	7/34 (20.6)	6/54 (11.1)	0.22
Comorbidities				
Diabetes mellitus	13/88 (14.9)	4/34 (11.8)	9/54 (16.7)	0.53
Chronic kidney disease	9/88 (10.2)	4/34 (11.8)	5/54 (9.3)	0.73
Liver disease	2/88 (2.3)	1/34 (2.9)	1/54 (1.9)	1.00
Pulmonary disease	44/88 (50)	19/34 (55.9)	25/54 (46.3)	0.38
COPD	11/44 (25)	5/19 (26.3)	6/25 (24)	1.00
Bronchitis	6/44 (13.6)	1/19 (5.3)	5/25 (20)	0.21
Emphysema	1/44 (2.3)	0 (0)	1/25 (4)	1.00
CF	2/44 (4.5)	1/19 (5.3)	1/25 (4)	1.00
Lung cancer	2/44 (4.5)	1/19 (5.3)	1/25 (4)	1.00
Non-CF Bronchiectasis	14/44 (31.8)	5/19 (26.3)	9/25 (36)	0.50
Asthma	9/44 (20.5)	7/19 (36.8)	2/25 (8)	**0.02**
Interstitial lung disease	1/44 (2.3)	1/19 (5.3)	0 (0)	0.43
Nodules/Masses	3/44 (6.8)	2/19 (10.5)	1/25 (4)	0.57
Empyema	1/44 (2.3)	1/19 (5.3)	0 (0)	0.43
Fibrosis	1/44 (2.3)	0 (0)	1/25 (4)	1.00
Immunocompromised ^a^	20/88 (22.7)	11/34 (32.4)	9/54 (16.7)	0.09
Prior NTM infection	15/88 (17)	5/34 (14.7)	10/54 (18.5)	0.64
Treated	9/15 (60)	4/5 (80)	5/10 (50)	0.58
Not treated	6/15 (40)	1/5 (20)	5/10 (50)	0.58
Prior MTB treatment	4/88 (4.5)	4/34 (11.8)	0 (0)	**0.02**
Smoking	42/88 (47.7)	20/34 (58.8)	22/54 (40.7)	0.10

Significant *p*-values < 0.05 indicated in bold; Data are presented as *n* (%); ^a^ Includes malignancy, history of prior transplant, and human immunodeficiency virus (HIV); Abbreviations: COPD, chronic obstructive pulmonary disease; CF, cystic fibrosis; MTB, *Mycobacterium* tuberculosis; NTM, nontuberculous mycobacteria.

**Table 5 antibiotics-11-01720-t005:** Safety and Tolerability of Antimicrobials in the Treatment Outcome Cohort.

	All (*n* = 88)	Unfavorable (*n* = 34)	Favorable (*n* = 54)	*p*-Value (95% Cl)
Antimicrobial Change, *n* (%)	53/88 (60.2)	25/34 (73.5)	28/54 (51.9)	**0.043**
1 only	15/53 (28.3)	5/25 (20)	10/28 (35.7)	0.21
≥2	38/53 (71.7)	20/25 (80)	18/28 (64.3)	0.21
≥3	29/53 (54.7)	15/25 (60)	14/28 (50)	0.47
≥4	21/53 (39.6)	13/25 (52)	8/28 (28.6)	0.082
≥5	14/53 (26.4)	10/25 (40)	4/28 (14.3)	**0.034**
Reasons for Change				
Drug-Drug Interaction	2/53 (3.8)	1/25 (4)	1/28 (3.6)	1.00
Susceptibility	13/53 (24.5)	5/25 (20)	8/28 (28.6)	0.47
Disease Status	14/53 (26.4)	10/25 (40)	4/28 (14.3)	**0.034**
Escalation	12/53 (22.6)	9/25 (36)	3/28 (10.7)	0.51
De-escalation	5/53 (9.4)	3/25 (12)	2/28 (7.1)	0.58
Treatment Optimization	9/53 (17)	4/25 (16)	5/28 (17.9)	1.00
ADEs	36/53 (67.9)	16/25 (64)	20/28 (71.4)	0.56
Drug Allergy	6/36 (16.7)	3/16 (18.8)	3/20 (15)	1.00
AKI	1/36 (2.8)	1/16 (6.3)	0 (0)	0.44
DILI	3/36 (8.3)	3/16 (18.8)	0 (0)	0.08
GI Intolerance	14/36 (38.9)	5/16 (31.3)	9/20 (45)	0.40
General Intolerance	10/36 (27.8)	7/16 (43.8)	3/20 (15)	0.07
Other	18/36 (50)	9/16 (56.3) ^a^	9/20 (45) ^b^	0.50
Drug Access	6/53 (11.3)	3/25 (12)	3/28 (10.7)	1.00
Insurance	5/6 (83.3)	3/3 (100)	2/3 (66.7)	1.00
National back order	1/6 (16.7)	0 (0)	1/3 (33.3)	1.00
Administration Issue	7/53 (13.2)	5/25 (20)	2/28 (7.1)	0.23
NPO	2/7 (28.6)	2/5 (40)	0 (0)	1.00
No IV access	3/7 (42.9)	2/5 (40)	1/2 (50)	1.00
PO regimen only	2/7 (28.6)	1/5 (20)	1/2 (50)	1.00
Dialysis access	1/7 (14.3)	1/5 (20)	0 (0)	1.00
Other	8/53 (15.1)	6/25 (24)	2/28 (7.1)	0.13
Nonspecific/Unclear	12/53 (22.6)	7/25 (28)	5/28 (17.9)	0.38

Significant *p*-values < 0.05 indicated in bold; ^a^ CNS side effects (*n* = 3), Cytopenia (*n* = 1), Cardiac side effects (*n* = 1), Tinnitus (*n* = 2), Visual disturbances (*n* = 2); ^b^ CNS side effects (*n* = 1), Hearing loss (*n* = 1), Infusion-related reaction (*n* = 1), Linezolid-related side effects (*n* = 1), Loss of voice (*n* = 1), Muscle aches (*n* = 1), Cardiac side effects (*n* = 1), Tinnitus (*n* = 1), Visual disturbances (*n* = 1); Abbreviations: ADEs, adverse drug events; AKI, acute kidney injury; DILI, drug-induced liver injury; GI, gastrointestinal; NPO, nothing by mouth; IV, intravenous; PO, by mouth.

**Table 6 antibiotics-11-01720-t006:** Univariable and Multivariable Regression Analysis for Predicting a Favorable Outcome.

Univariable Analysis	Odds Ratio	95% Cl	*p*-Value
Greater than 5 Antimicrobial Changes	0.192	0.055–0.675	0.010
Private Insurance	6.150	1.310–28.930	0.021
Underweight	0.192	0.055–0.675	0.010
History of Asthma	0.148	0.029–0.764	0.022
Antimicrobial Change, Intolerance	0.249	0.065–0.962	0.044
**Multivariable Analysis ^a^**			
Greater than 5 Antimicrobial Changes	0.187	0.049–0.714	0.014
Private Insurance	6.112	1.12–33.29	0.036

^a^ Multivariable analysis was adjusted for history of asthma, history of treatment of prior MTB, and general intolerance.

## Data Availability

The data presented in this study are available on request from the corresponding author. The data are not publicly available due to institutional policy.

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
