# Peer review of "Epidemiology, Outcomes and Tolerability of Protracted Treatment of Nontuberculous Mycobacterial Infections at a Community Teaching Hospital in the Southeastern United States"

_antibiotics, 2022, doi:10.3390/antibiotics11121720_

Round 1

Reviewer 1 Report

This is a well written article about 10 years' experience with a large cohort of NTM infections at a single center. 

Would recommend defining unfavorable outcomes. Also mention the average duration of follow up for these patients. Comment on any other non-culture methods used in the patients for identifying NTM infections.

Author Response

We appreciated reviewer 1's feedback. The following clarification to study definition was incorporated: unfavorable treatment outcome was defined was mortality or transitioned to palliative or hospice care while on therapy, or termination of treatment due to antimicrobial intolerability or lack of clinical improvement (line 256-258).

Furthermore, only culture method was utilized to retrospectively identify eligible patients for study inclusion.

Reviewer 2 Report

The manuscript titled “Epidemiology, Outcomes, and Tolerability of Protracted Treatment of
Nontuberculous Mycobacterial Infections at a Community Teaching Hospital in the Southeastern United States” analyzed the NTM infection cases between 2010 and 2020 in a hospital located in the southeast US and provided some valuable insight on the factors that would affect the treatment outcomes.

The method design is clear and reasonable. The evidence would be stronger if the sample size is bigger.

The authors mentioned in the introduction that current guidelines were based on data reported outside of the US. Did the author ever compare their results with the reported data? It would be interesting to see the differences or agreements between the current data and previous data.

Author Response

We thank reviewer 2 for their feedback in regards to comparing previous data to our current study findings. This comparison was made in the discussion to highlight similar findings in the existing data in terms of distribution pattern of NTM species and the susceptibility profile for MAC vs M. abscessus; whereas some minor differences were observed for rates of treatment success, but an overall similarity in terms for rates of antimicrobial associated ADEs. 

Reviewer 3 Report

1.      What is the main question addressed by the research?

The authors performed a retrospective, observational study including a large cohort of patients with NTM during a 10-years period.

The authors addressed some important issues in infectious diseases: epidemiology of NTM, risk factors for unfavorable outcome, antimicrobial susceptibility of Mycobacteria and reasons of change in antimicrobial therapy.

2.      Do you consider the topic original or relevant in the field? Does it address a specific gap in the field?

This is a relevant topic in the field because it concern immunodepressed patients and the knowledge of My susceptibility to different antibiotics,  which is variable and depends on specific individual and regional factors, could help improving early management of such diseases.

3. What does it add to the subject area compared with other published material?

The article brings as a novelty a detailed analysis of the sensitivity to antibiocrobials for each type of  isolated Mycobacteria  and for the type of their growth (slow or rapid) in a specific geographic area: southeastern US.

4. What specific improvements should the authors consider regarding the

methodology? What further controls should be considered?

-

5. Are the conclusions consistent with the evidence and arguments presented

and do they address the main question posed?

 The conclusions are consistent with the evidence and analysis and they address the main question posed

6. Are the references appropriate? Yes

7. Please include any additional comments on the tables and figures.-

The data in tables and figures are interpreted appropriately and consistently throughout the manuscript.

I have some Minor concerns:

Some statements should be rephrased thus, what was previously put as:

“79  Overall, amikacin was the only antimicrobial that remained highly susceptible to all the rapid growers

 84 Contrary to the rapid growers, multiple antimicrobials remained highly susceptible to MAC 85 isolates including.

 156. The high susceptibility of clarithromycin and amikacin for MAC is reassuring”

Considering that Bacteria are susceptible not antibiotics

Author Response

We thank reviewer 3's feedback. The following statements where rephrased: (1) the current finding of MAC isolates remained highly susceptible to clarithromycin and amikacin (line 150-151), (2) all the rapid growers remained highly susceptible to amikacin (line 79), and (3) MAC isolates remained highly susceptible to multiple antimicrobials (line 84).